# Efficient magnetic switching in a correlated spin glass

Juraj Krempaský [1] ✉, Gunther Springholz[2], Sunil Wilfred D'Souza[3], Ondřej Caha[4], Martin Gmitra [5,6], Andreas Ney [2], C. A. F. Vaz [1], Cinthia Piamonteze [1], Mauro Fanciulli [7], Dominik Kriegner [8,9], Jonas A. Krieger[10,12], Thomas Prokscha [10], Zaher Salman [10], Jan Minár [3] ✉ & J. Hugo Dil [1,11] ✉

The interplay between spin-orbit interaction and magnetic order is one of the most active research fields in condensed matter physics and drives the search for materials with novel, and tunable, magnetic and spin properties. Here we report on a variety of unique and unexpected observations in thin multiferroic $Ge_{1-x}Mn_xTe$ films. The ferrimagnetic order parameter in this ferroelectric semiconductor is found to switch direction under magnetostochastic resonance with current pulses many orders of magnitude lower as for typical spin-orbit torque systems. Upon a switching event, the magnetic order spreads coherently and collectively over macroscopic distances through a correlated spin-glass state. Utilizing these observations, we apply a novel methodology to controllably harness this stochastic magnetization dynamics.

Recently, combining semiconducting and magnetic properties within the same material changed from being a concept[1] to being one of the main progress vectors for spintronics[2], in particular when combining magnetism with topological properties[3]. With the discovery of ferroelectric Rashba semiconductors (FERS)[4], magnetic doping in dilute magnetic semiconductors (DMS) has opened pathways to exploit the electron spin associated with the Rashba-Zeeman type spin splitting of the electronic states[5–7]. To date the primary FERS representative is $\alpha$-GeTe; with only two atoms per unit cell and ≈0.3 Å displacement along the ⟨111⟩ direction between the Ge and Te atoms, it is arguably the simplest room temperature ferroelectric semiconductor[8,9]. The combination of ferroelectric order and large spin-orbit interaction (SOI) yields a switchable Rashba-type spin structure of the bulk states[10], whereby the states are fully spin polarised around the valence band maximum[11]. When doped with Mn, magnetic order is induced in $Ge_{1-x}Mn_xTe$ while the ferroelectric order remains for dopings below $x = 25\%$, rendering it a multiferroic semiconductor with broken inversion symmetry[6,12–14]. Moreover, the collinear alignment of the magnetisation and ferroelectric polarisation axes in $Ge_{0.87}Mn_{0.13}Te$ ensures magnetoelectric coupling in the system[10]. Conversely, as there is now a cohesive picture between Berry curvature and emergence of anomalous Hall effect in non-centrosymmetric magnetic semiconductors[15], the current-induced magnetization switching mechanism in this material points to a bulk Rashba-Edelstein effect[7]. Both spectroscopic[6,10] and transport studies[7] indicate that the key ingredient for the magnetoelectric functionality of $Ge_{1-x}Mn_xTe$ is its carrier-induced magnetism[16–20] mediated by the valence band[6,7]. Furthermore, multiferroic $Ge_{1-x}Mn_xTe$ is proposed to be a unique platform to explore novel phenomena such as nonreciprocal electric transport[21] or magnetic skyrmionic polarons[22].

¹Photon Science Division, Paul Scherrer Institut, CH-5232 Villigen, Switzerland. ²Institut für Halbleiter-und Festkörperphysik, Johannes Kepler Universität, A-4040 Linz, Austria. ³New Technologies-Research Center University of West Bohemia, Plzeň, Czech Republic. ⁴Dept. of Condensed Matter Physics, Masaryk University, Kotlářská 267/2, 61137 Brno, Czech Republic. ⁵Institute of Physics, P. J. Šafárik University in Košice, Park Angelinum 9, 040 01 Košice, Slovakia. ⁶Institute of Experimental Physics, Slovak Academy of Sciences, Watsonova 47, 040 01 Košice, Slovakia. ⁷LPMS, CY Cergy Paris Université, 95031 Cergy-Pontoise, France. ⁸Institute of Physics ASCR, v.v.i., Cukrovarnická 10, 162 53 Praha 6, Czech Republic. ⁹Dept. of Condensed Matter Physics, Charles University, Ke Karlovu 5, 121 16 Praha 2, Czech Republic. ¹⁰Laboratory for Muon Spin Spectroscopy, Paul Scherrer Institute, CH-5232 Villigen PSI, Switzerland. ¹¹Institut de Physique, École Polytechnique Fédérale de Lausanne, CH-1015 Lausanne, Switzerland. ¹²Present address: Max Planck Institut für Mikrostrukturphysik, Weinberg 2, 06120 Halle, Germany. ✉e-mail: juraj.krempasky@psi.ch; jminar@ntc.zcu.cz; hugo.dil@epfl.ch

In this work, we combine a range of spectroscopic techniques based on X-ray magnetic circular dichroism (XMCD), near-edge X-ray absorption fine structure spectroscopy (NEXAFS) and state of the art modeling (see Methods I–IV), to reveal that $Ge_{0.87}Mn_{0.13}Te$ is a bulk ferrimagnet (FiM). This is corroborated by complementary low-energy muon spectroscopy (LE-$\mu$SR) as a local probe to characterize the temperature-dependent magnetic transitions on the nanometer scale and SQUID magnetometry to access the macroscopic magnetisation. The results testify that the FiM order builds up below $T_{c1} = 100$ K on top of a paramagnetic (PM) background and that below $T_{c2} = 40$ K the system behaves as a spin-glass. By developing new analysis tools based on magnetostochastic resonance (MSR) switching we address the $Ge_{0.87}Mn_{0.13}Te$ magnetism in the static and dynamical regimes. The

ensemble of experimental techniques and theory yields a holistic view of a correlated spin glass with magnetization switching that can be driven by nanoampere pulsed currents using stochastic resonance. In particular we show that this switching is collective due to glassiness, and that the magnetic switching can be turned on and off by small changes in pulse periodicity.

## Results

### Ferrimagnetic switching

Figure 1a shows XMCD spectra measured at 10 K in total electron yield (TEY) mode while applying magnetic fields up to ±6 T. As will be discussed later, the dichroic signal does not saturate at our maximum B-fields due to a paramagnetic spin-glass state, similar to the

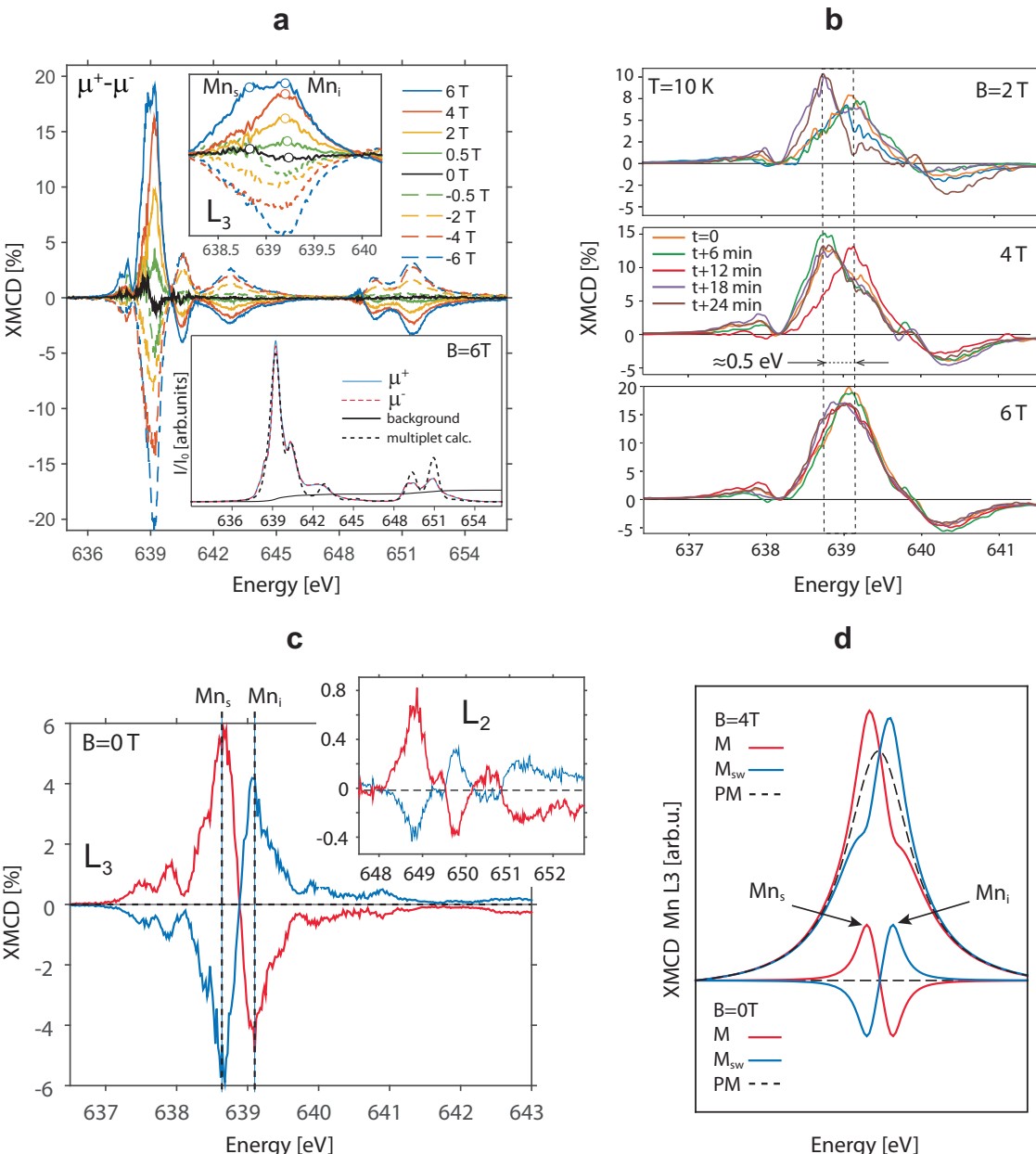

**Fig. 1 | Switching of ferrimagnetic order in XMCD. a** XMCD spectra for selected applied B-fields. Top inset: zoom into the Mn $L_3$ edge, showing two distinct spectral weight from substitutional $Mn_s$ and interstitial $Mn_i$ atoms. Bottom inset: background subtracted XAS spectra for $\pm\mu$ light polarizations, measured at $B = 6$ T (10 K), and overlaid with multiplet calculations. **b** Series of Mn$_{L_3}$ XMCD spectra taken at ≈6 min intervals for selected B-fields. **c** Spontaneous switching recorded in two

consecutive XMCD datasets (red/blue) in the absence of applied B-field, shown at the Mn$_{L_3}$ (main graph) and Mn$_{L_2}$ (inset) absorption edges. **d** Mn$_{L_3}$-edge model based on 4 Gaussian functions simulating $Mn_s$ and $Mn_i$ spectral contributions. For PM the $Mn_s$ and $Mn_i$ features are aligned, with their amplitude dependent on the B-field. The FiM consists of oppositely aligned peaks with different amplitude. $M$ and $M_{SW}$ refer to the reference and switched state respectively.

$Cd_{1-x}Mn_xTe$ DMS system[23]. The top-inset is a zoom-in into the $Mn_{L3}$ absorption edge, showing two distinct XMCD features separated by ≈0.5 eV. A time series of XMCD spectra for selected B-fields are shown in Fig. 1b, revealing that these features fluctuate, eventually completely and spontaneously reversing at zero B-field (Fig. 1c). The shape of the dichroic signal at the $Mn_{L3}$-edge can be understood from the combination of a paramagnetic background superposed with ferrimagnetic contributions (Fig. 1d). The fluctuating spectral weight in Fig. 1b indicates that the switching continues for applied fields even above 4 T. From detailed analysis of an extensive data set obtained from several experimental runs and different samples, we find that the switching between the two magnetic orientations appears to be regular and does not show the telegraph noise expected for a system which randomly switches between two magnetic- or spin-states[24,25]. As surprising as the switching itself, is the fact that it occurs simultaneously, within the time resolution of the experiment, over the square millimetre area probed by the photon beam spot.

To reiterate, the dichroism at $Mn_{L3}$ systematically shows two opposite magnetic moments from different types of Mn atoms, with absorptions at $hv = 638.8$ and 639.3 eV, simultaneously switching after about every energy scan across the $Mn_{L3}$ absorption edge, in the absence or presence of an externally applied field. Notably, the opposite magnetic moments are not exactly equal in magnitude, indicating a non-balanced ferrimagnetic order from two distinct Mn-sites. The spontaneous switching is further corroborated by the fact that we obtain an equivalent dichroism effect by measuring spectra with the same circular polarization (see Supplementary Fig. 1a). This also indicates that the light helicity is not the driving force for the switching. At this point, the following questions arise:

- What is the difference between the two Mn-sites?
- Why are their magnetic moments spontaneously reversing in time over macroscopic length scales?
- What is the microscopic origin of the magnetization switching?
- How to control the switching?

To address these questions, we first identify the different Mn contributions with our theoretical calculations, and then proceed to explain the spontaneous magnetic switching in terms of magneto-stochastic resonance, whereby periodic time fluctuations in the current associated with photoexcited electrons interfere with large TEY pulses during the XMCD measurements to yield a global resonant magneto-stochastic switching of the magnetisation. Finally, we show that the collective behavior originates from a glassy spin state.

## Magnetic ground state properties

In accordance with our earlier resonant angle-resolved photoemission (ARPES) studies on $Ge_{1-x}Mn_xTe$[6] and further XMCD studies on other Mn-doped DMS systems[26–28], we identify the magnetic moments in $Ge_{1-x}Mn_xTe$ to originate from substitutional ($Mn_s$) and interstitial ($Mn_i$) atoms. However, in contrast to other related systems, we find that in $Ge_{0.87}Mn_{0.13}Te$ the characteristic ≈0.5 eV energy separation between $Mn_s$ and $Mn_i$ is not due to Mn segregation or non-magnetic surface Mn-oxides, nor to a depletion of Mn in the near-surface layer. Instead, the energy shift is an intrinsic bulk property that we also see in bulk-sensitive fluorescence yield and in NEXAFS analysis from the Mn-K edge (see Supplementary Fig. 1b, c, respectively). From these data, we estimate the presence of $Mn_s$ and $Mn_i$ atoms in the GeTe host lattice, with a $Mn_s$:$Mn_i$ occupancy close to 2:1. Based on this experimental finding we calculate the $Ge_{0.87}Mn_{0.13}Te$ magnetic ground state properties using density functional theory, where the Mn-disorder is implemented within the coherent potential approximation (CPA) alloy theory (see Methods).

Figure 2 describes how the distinct Mn incorporation into the GeTe lattice induces magnetic frustration due to the various competing exchange interactions, while the ferroelectric crystal structure is retained. For 13% Mn-doping we expect nearly 1 Mn per 8 unit cells as schematically depicted in Fig. 2a. Our CPA calculations are based on an "infinite" $Ge_{0.87}Mn_{0.13}Te$ lattice consisting of 8.9% probability occupation of $Mn_s$ substituted on Ge lattice sites ($Mn_1$) and two types of interstitial $Mn_i$ atoms, $Mn_2$ and $Mn_3$, each one with 2.2% probability occupation. The most prominent contribution to the magnetic order comes from the Heisenberg exchange energy between $Mn_{2,3}$ and between $Mn_{1,2/3}$ atoms, whereas the interaction between substitutional atoms ($Mn_{1,1}$) is damped due to the presence of the interstitial Mn. The corresponding magnetic exchange couplings $J_{i,j}$ are presented in Fig. 2b as a function of the distance $R_{i,j}/a$ between atoms, where $a$ is the lattice parameter and $i, j = 1, 2, 3$. The positive exchange constants ($J_{i,j} > 0$) favor ferromagnetic (FM) order, whereas negative values ($J_{i,j} < 0$) favor antiferromagnetic (AFM) order (see Methods).

The co-existence of FM and AFM order in a DMS system is one of the main ingredients for magnetic frustration in a canonical spin glass, which in our case is further enhanced by Dzyaloshinskii-Moriya interaction (DMI). DMI is characteristic to non-centrosymmetric systems with large spin-orbit coupling (SOC) as clearly applies for $Ge_{1-x}Mn_xTe$, where it promotes non-coplanar (canted) arrangements between spin states. As shown by Fig. 2c, DMI is highly anisotropic in both inter- and intra-Mn interactions, amongst which $x2, 3$, $y2, 3$ and $z2, 2$ appear as most prominent (see Methods). Both DMI and all $J_{i,j}$ have fluctuating $R_{i,j}/a$ dependences. Furthermore, the DMI is non-negligible compared to the $J_{i,j}$ especially for larger length scales, leading to magnetic frustration at different length scales.

As shown in the inset of Fig. 2b, for short length scales there is a competition between FM and AFM contributions, the relative magnitudes of which explain why $Ge_{0.87}Mn_{0.13}Te$ stabilizes in a ferrimagnetic ground state. Since CPA allows us to derive spectral contributions from individual Mn-atoms (Fig. 2d), the resulting theoretical magnetic dichroism reproduces the measured spectra very well if we take into account that the majority of $Mn_s$ contribute to the PM background once the magnetic interactions stabilize, as detailed in Supplementary Fig. 2.

The described ferrimagnetic ground state, based on three distinct interacting Mn lattice sites, has been elusive in previous theoretical[20,29] and experimental studies[16,18,30] because only substitutional $Mn_s$-sites were considered (empty markers in Fig. 2b). The oscillatory long-range $J_{i,j}$ exchange of the $Mn_s$-sites is damped due to the presence of interstitial $Mn_i$ atoms (filled green markers in Fig. 2b). Furthermore, when probed under applied B-field (Fig. 1a), the opposite FiM contributions are placed on a large PM background due to which the $Mn_i$-moments partially align with those of $Mn_s$. Only the magnetization switching dynamics shown in Fig. 1b allows us to distinguish the two sites under applied magnetic field.

In order to assess the energetics of the FiM switching, we compare in Fig. 3a the energy scale of the rotation of $Mn_i$ spins with respect to the $Mn_s$ spin held fixed along the [111] axis by the magnetocrystalline anisotropy energy (MAE, see Methods). The combination of both energy profiles constitutes a double-well uniaxial potential with a MAE barrier of ≈0.1 meV, comparable to that of ferromagnetic $3d$ metals[31]. Having identified the general magnetic order and its bistable potential model, we now turn to the dynamics and the unexpected magnetic switching.

## Magnetostochastic resonance switching

The dynamical spontaneous magnetisation reversal in our system is explained by the mechanism of stochastic resonance (SR). Generally speaking, SR is a phenomenon applicable to nonlinear systems whereby a weak signal is amplified through an entraining periodic signal or even noise. Besides the bistable potential, SR requires two additional ingredients[32]: (i) a weak periodic input and (ii) a source of noise, or amplitude modulated signal[33] entrained with the periodic input. As will be elaborated in detail below, the first ingredient is provided by periodic TEY oscillations $P(t)$ upon $Ge_{0.87}Mn_{0.13}Te$ x-ray

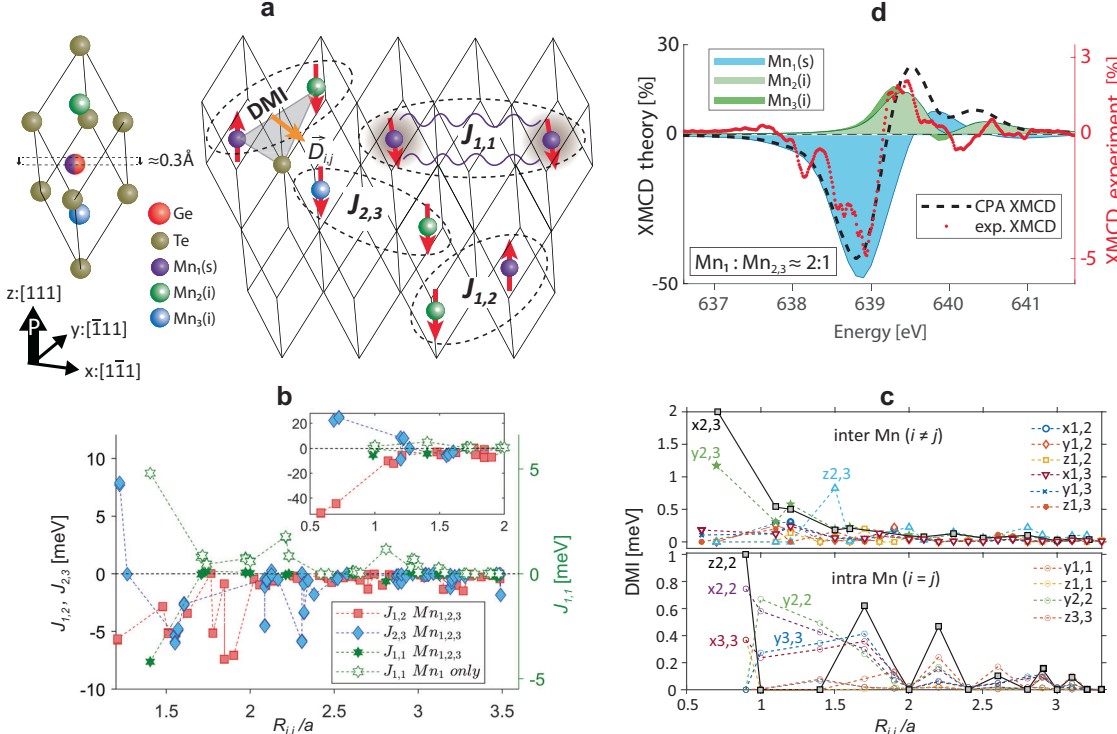

**Fig. 2 | Frustrated magnetic interactions in Ge$_{0.87}$Mn$_{0.13}$Te. a** Illustration of Mn doping in the rhombohedrally distorted α-GeTe unit cell with one substitutional Mn$_s$ and two potential interstitial Mn$_i$ dopants. **b** Exchange couplings $J_{i,j}$, between Mn$_i$-Mn$_j$ atoms with all three types of Mn atoms present (full markers), and with only substitutional Mn$_1$ atoms (empty marker). Distances are normalized to the lattice parameter $a$. Inset: Extension to smaller atomic distances. **c** DMI values for inter- and intra-Mn site coupling. The $D_{i,j,x}$ component is summarized as $xi,j$ and similar for $y$ and $z$ ($x, y, z$ index directions are denoted in **a**). Dominant interactions are indicated in the graph, others listed in legends. **d** Comparison between XMCD experiment (red dotted line) and CPA theory, in which spectral contributions from individual Mn atoms result in a final XMCD signal (dashed line).

illumination, whereas the second $\eta(t)$ originates from the short TEY pulse in x-ray absorption during the scan over the Mn$_{L3}$ absorption edge, both subjected to thermal noise. The sum of these induce the observed transition between the two FiM equilibria as depicted in Fig. 3a, b.

The periodic drive $P(t)$ represents a continuous sinusoidal signal with $T_\Omega \approx 5$ min (Fig. 3c). These systematically observed oscillations are present also while performing $h\nu$ scans; they are not related to the periodic top-up current refill of the synchrotron and do not depend on x-ray polarization and applied B-field (see Supplementary Fig. 3). Therefore, our understanding is that they originate from charging and discharging inside a capacitor circuit with Ge$_{0.87}$Mn$_{0.13}$Te as dielectric, whereby the primary charging effect is caused by charge separation due to a steady-state bulk photovoltaic effect typical to materials with broken inversion symmetry[34]. For the GeTe host material this shift current effect is even stronger compared to conventional oxide ferroelectrics[35]. Moreover, the system possesses self-poling properties[36] and negative capacitance[37], leading to thermodynamically unstable transient effects[38].

The photoelectrons contributing to the TEY current leave the sample following the shortest path. On average this is along the normal to the sample plane, i.e. along the (Ge,Mn)Te [111] direction. This current direction is collinear with both the perpendicular magnetic anisotropy, as well as the ferroelectric polarisation. In line with the exertion of SOT by the inverse Rashba-Edelstein effect[7], the SOT-induced switching in our case is triggered by the TEY current, especially the short pulse $\eta(t)$ at the Mn$_{L3}$-edge (Fig. 3c). In fact, in the dichroic signal we never observe anomalous or discontinuous spectral shapes, indicating that the switching event most likely occurs directly after the Mn$_{L3}$-edge.

## Harnessing the magnetostochastic resonance switching

The phenomenological description of magnetostochastic resonance (MSR) applied to magnetization dynamics typically predicts a million times higher switching probability for bistable magnetic systems[39]. Consistent with these predictions, even though the geometry and time scales are different, for Ge$_{0.87}$Mn$_{0.13}$Te the current-driven magnetisation switching in the milliampere regime[7] is achieved under MSR with a nanoampere TEY pulse at the Mn$_{L3}$ absorption edge, and thus with unprecedented efficiency of low-current density pulses compared to typical spin-orbit torque systems. The TEY current is typically in the 10-100 nA range. In ref. 7 the typical current density used to switch the magnetization is $\approx 6 \times 10^6$ Acm$^{-2}$. This huge current density is applied within a pulse current of 1 ms. On the other hand, in our case the TEY current from a sample area of $\approx 1$ mm$^2$ amounts to $100 \times 10^{-9}/(0.1 \times 0.1) \approx 1 \times 10^5$ Acm$^{-2}$, within a period of time during the on-the-fly energy scan[40] across the Mn L3-edge (ca.1–2 s). Considering the pulse duration, a coarse estimate between the two experiments yields a current density ratio of $6 \times 10^6/(2000 * 10^{-5}) \approx 3 \times 10^8$. This indicates that for the magnetic switching, the current density per unit time is in our case >6 orders of magnitude lower compared to ref. 7. It should be noted that in conventional slow point-by-point scans the FiM switching is not observed. This underlines the relevance of dynamics in the magnetostochastic resonance switching.

A typical measurement series is represented by the green top traces in Fig. 3d, where the resulting TEY signal is a two-frequency system in which $P(t)$ realizes frequency modulation and $\eta(t)$ implements amplitude modulation as a deterministic additive noise. These two driving frequencies, schematically depicted in Fig. 3c, exemplify stochastic resonance in two-frequency signal systems[33]. With such a combination of periodic and aperiodic driving frequencies the SR can

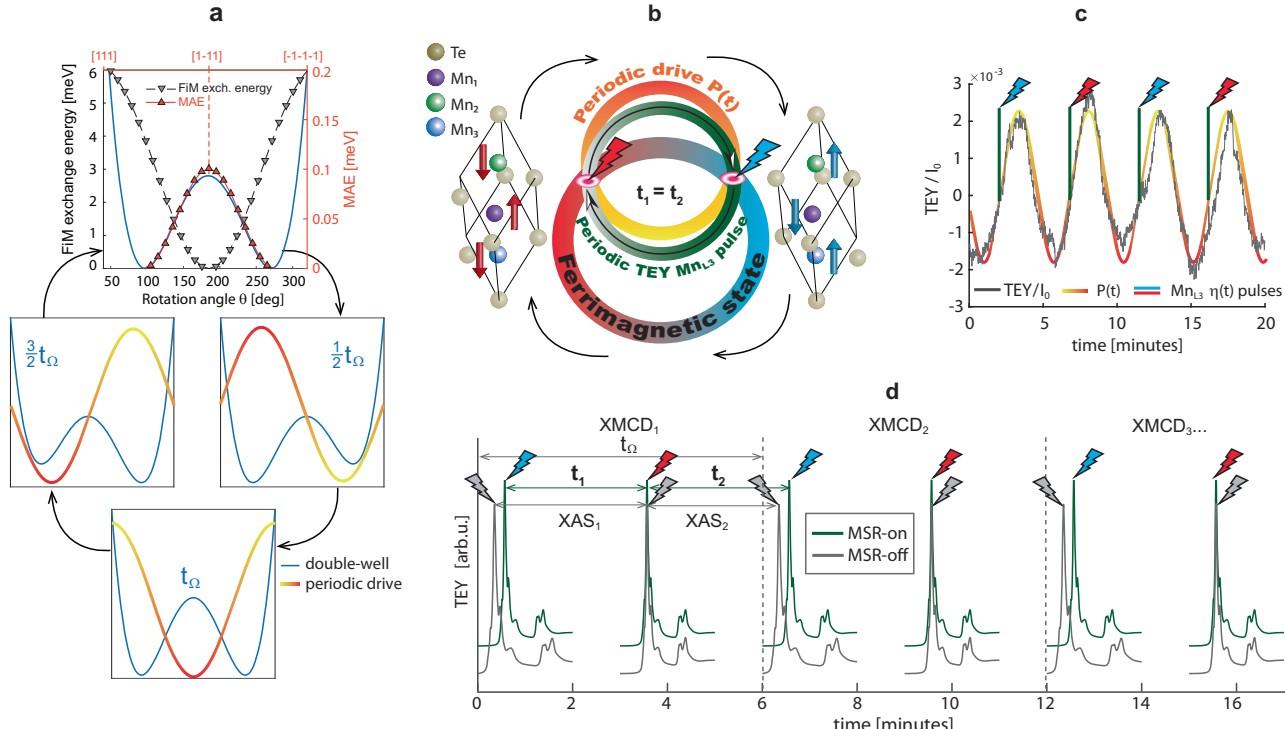

**Fig. 3 | Magnetostochastic resonance driving sources. a** Stochastic resonance switching on a symmetric double-well qualitatively modelled by FiM exchange energy (grey downward triangles) and magnetocrystalline anisotropy energy (red upward triangles). The potential is subject to a synchronised periodic drive $T_\Omega$ (red-yellow trace), enabling the switching of the otherwise stable FiM state. **b** FiM switching cartoon: the circles illustrate the MSR harnessing concept with periodic $Mn_{L3}$ TEY pulses indicated by grey and red/blue flash symbols, respectively. Periodic pulses with entrained $P(t)$ signal (red-yellow circle) enable the FiM switching in a statistical average. **c** Sinusoidal fit (red-yellow trace) of a time evolution of normalised TEY periodic drive $P(t)$ measured at 10 K, $B = 0$, and $h\nu = 638.8\,eV$ (grey trace). Superimposed red and blue spikes represent transient TEY pulse at the $Mn_{L3}$ edge that would occur during consecutive $XAS_1$, $XAS_2$, etc. scans. **d** Typical time series of periodic (green/top) and aperiodic (grey/bottom) XAS scans which controls the magnetostochastic resonance switching (MSR-on/off). Empty spaces between the XAS scans is the time needed to move the monochromator back to the initial energy.

be harnessed, e.g. controllably suppressed by detuning the frequencies from their optimal setting, as schematically depicted by aperiodic gray flash symbols in Fig. 3d. We achieved this by measuring XMCD data with different photon energy ranges (grey bottom traces in 3d), thereby moving away from the optimal synchronization with driving frequency $t_1 = t_2 = \frac{1}{2}t_\Omega$[32], where $t_{1,2}$ is the time needed to measure one XAS spectrum, and $t_\Omega$ one full XMCD dataset (i.e. two XAS spectra measured with opposite circular polarisations).

Figure 4 summarizes the MSR harnessing with two synchronized loops with aperiodic (MSR-off) and periodic (MSR-on) additive noise $\eta(t)$. The data indicates that there is an immediate change from the MSR-off regime without FiM switching to FiM switching under the MSR-on regime. We emphasize that under MSR-on, the FiM switching holds only in the statistical average, quantified via the residence time distribution $N(t)$[32]. We quantified $N(t)$ by counting the multiples of $t_\Omega$ between subsequent FiM switching occurrences, leading to a sequence of Gaussian-like peaks with exponentially decreasing envelope as seen in Fig. 4d at 5 K. In qualitative agreement with SR predictions[32], the peak distributions under thermal noise shifts to shorter times, as seen at 10 K. The $N(t)$ distribution envelope is found to depend on the resonance conditions through the driving period $t_\Omega$ (see Supplementary Fig. 4a), which further confirms the MSR origin of the energy efficient FiM switching with drastically reduced currents.

The observed SR switching allows us to sharpen the microscopic understanding of the $Ge_{0.87}Mn_{0.13}Te$ magnetic order. To this end we analyze the orbital $m_L$ and spin $m_S$ contributions from sum rules[41,42]. Because the magnitude and sign of the MSR-on XMCD signal changes in time, the directional sense of the magnetic

moments with respect to the beam axis is changing as well, occasionally yielding a totally quenched moment $M_{tot} = m_S + m_L$ such as in the grey traces in Fig. 4c. To avoid uncertainties imposed by this, we therefore focus on the MSR-off regime for the B-field dependence of $m_L$ and $m_S$ contributions, as summarized in Fig. 5. The theoretically obtained $m_L$ of $-0.032\,\mu_B/$(Mn atom) is independent of the magnitude of the applied field and the total magnetisation $M_{tot}$ is expected to be dominated by $m_S$, which takes a value of $m_S \approx 2.1\,\mu_B/$(Mn atom) at saturation. This trend is well observed in Fig. 5a where the applied B-field drives $m_S$, whereas $m_L$ is rather constant. That the local Mn-moments deduced from the sum rules $(0.13 \pm 0.02\,\mu_B/$Mn atom$)$ for $B = 0$ differ from the theoretical values is due to the fact that most Mn-moments from the PM background do not contribute to the measured moments. Table 2 in the Methods section compares the calculated Mn-moments with experimental data in the MSR-off regime.

The macroscopic perspective of the magnetic order in $Ge_{0.87}Mn_{0.13}Te$ is given by SQUID magnetometry as shown in Fig. 5b. A clear hysteresis loop is observed with a narrow waist and low coercive field. A remarkable feature are the dip-peak structures around zero field indicated by arrows. Similar non-monotonic features were observed in the magnetic field dependence of the $Ge_{0.87}Mn_{0.13}Te$ anomalous Hall component, which were attributed to noncoplanar spin textures with scalar spin chirality[7]. This consideration is supported by the magnetic frustration due to different $J_{i,j}$ and DMI length scales summarized in Fig. 2c, d. Interestingly, the non-monotonic features in SQUID, appearing under decreasing applied B-field, suggest the possible formation of frustrated spin textures like magnetic skyrmions[43].

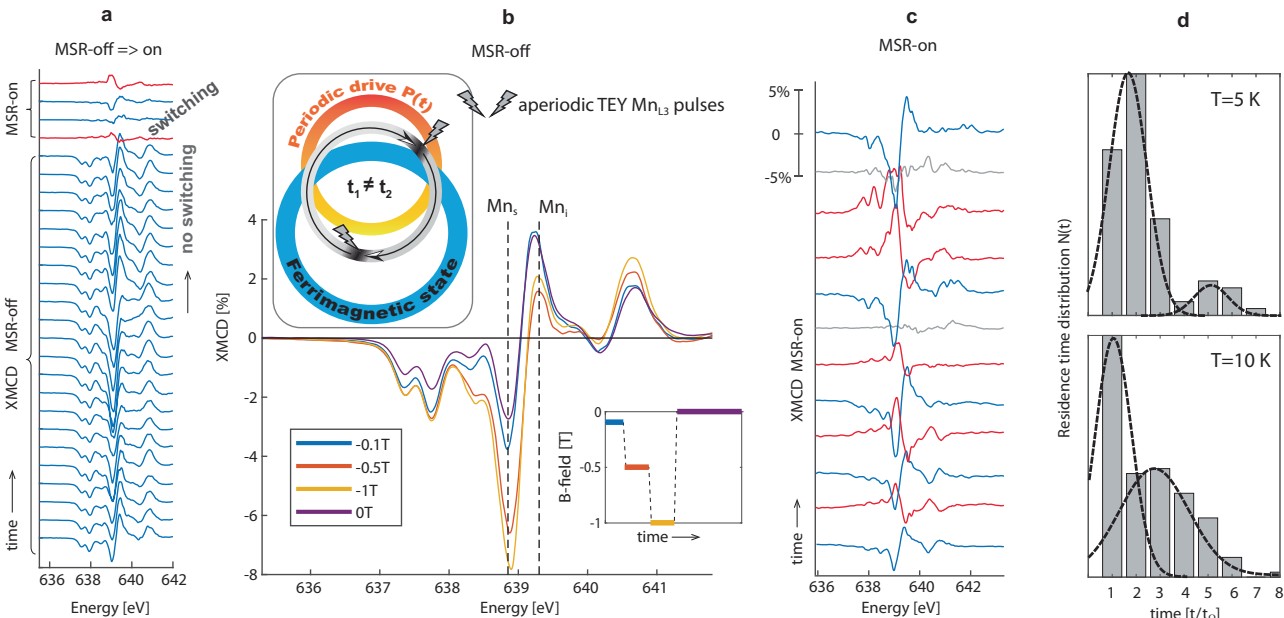

**Fig. 4 | Harnessing switching dynamics with magnetostochastic resonance.**
**a** Stacked plot of $Mn_{L3}$ XMCD spectra measured in MSR-off regime directly followed by MSR-on regime. **b** XMCD MSR-off spectra averaged for individual applied B-fields as depicted in the inset. Top cartoon illustrates that under aperiodic $Mn_{L3}$ TEY pulses the MSR switching is disabled. **c** Stacked plot of MSR-on XMCD spectra. In (**a**) and (**c**) blue/red traces reflect the original/switched FiM state, grey traces reflect totally quenched magnetic state. **d** Statistical FiM switching quantification by residence time distributions $N(t)$, fitted with Gaussian functions at 5 and 10 K.

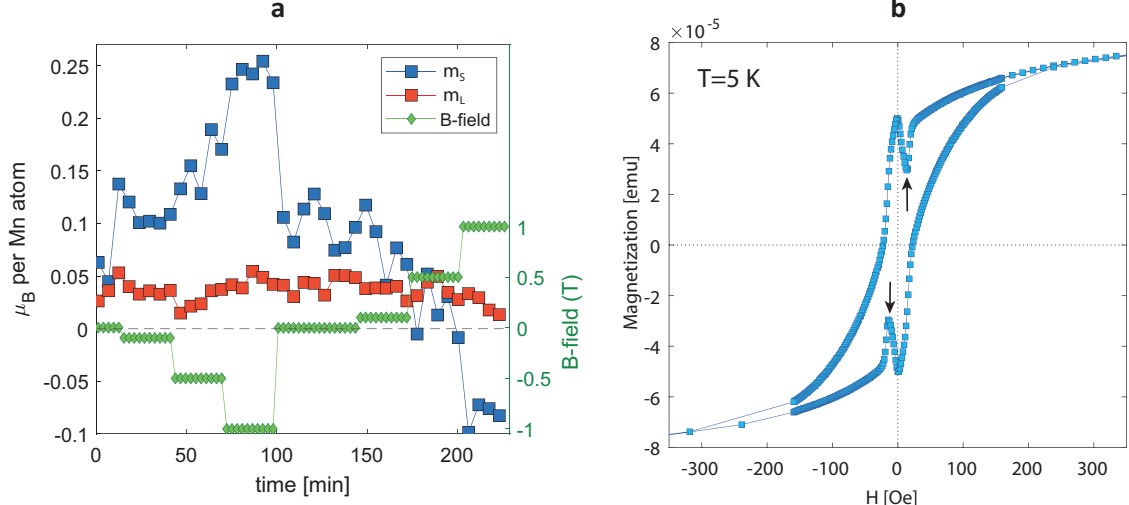

**Fig. 5 | Microscopic and macroscopic magnetisation in XMCD and SQUID. a** Time series of spin ($m_S$) and orbital ($m_L$) moments showing their dependence on applied B-field under the MSR-off conditions after zero-field cooling (ZFC). **b** Out-of-plane SQUID hysteresis for $Ge_{0.87}Mn_{0.13}Te$ grown on InP(111).

## Correlated spin glass

In addition, the temperature dependence of the magnetization shown in Fig. 6b does not follow the typical $(1 - (T/T_c)^\alpha)^\beta$ magnetization dependence with a single transition temperature, but rather a steadily diminishing magnetization with various critical temperatures indicated in the figure. The splitting between zero-field-cooled (ZFC) and field-cooled (FC) curves is a further clear indication of competing energy scales and spin orders, and of possible spin excitations in magnetic glassy-phases[44]. LE-$\mu$SR is the method of choice to characterize magnetically ordered volumes inside such a spin-glass from a local point of view.

In $Ge_{0.87}Mn_{0.13}Te$ we find that muons stopping in magnetic regions of the sample depolarize almost immediately and do not contribute to the measured signal. However, muons that land in paramagnetic regions experience predominantly the applied field ($\approx$5 mT). The polarization of these muons can be used to extract the three quantities presented in Fig. 6a, namely, (i) the local mean field $B$ sensed in the paramagnetic regions (top); (ii) the damping rate $\lambda$ of the oscillations which is proportional to the width of local field distribution (middle); and (iii) the volume fraction of the paramagnetic regions (bottom). All three parameters start to change at the onset of magnetic order ($\approx$100 K). Note that the paramagnetic (magnetically ordered) volume fraction decreases (increases) gradually at the onset of magnetic order, clearly showing the gradual nature of this transition. This is accompanied by a decrease in $B$, indicating that the magnetic regions have a net magnetic moment producing an increasing demagnetizing fields in the paramagnetic regions with decreasing temperatures. Below $\approx$40 K, most of the

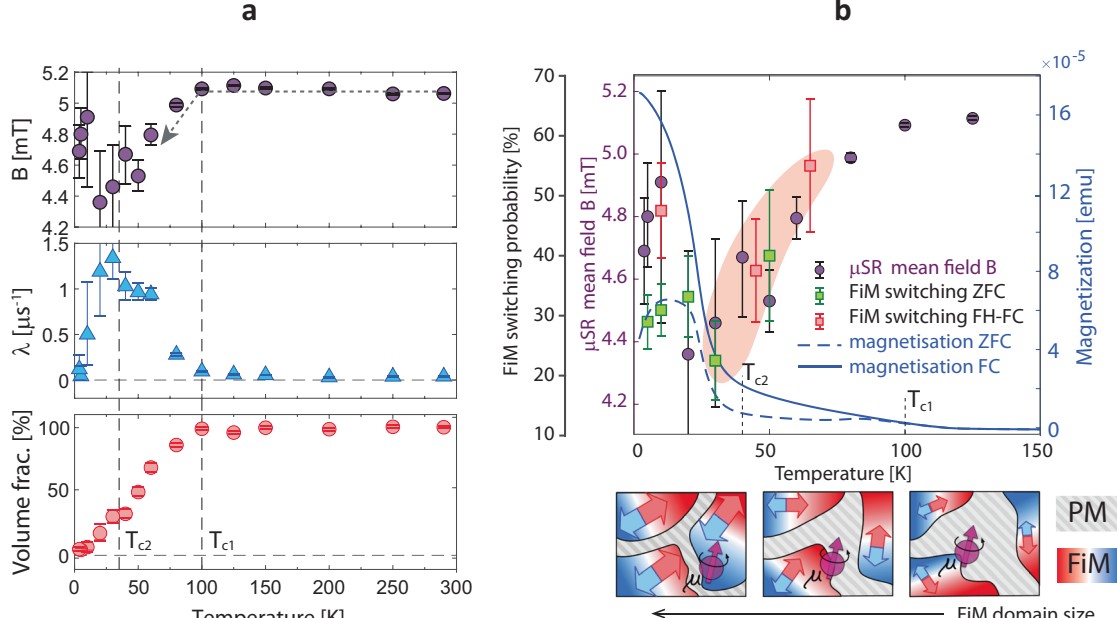

**Fig. 6 | Correlated spin glass state observed by LE-μSR, SQUID, and switching dynamics. a** Temperature-dependent average local field $B$ (top), muon damping rate $\lambda$ (middle), volume fraction of the paramagnetic regions (bottom), obtained from fits of the LE-μSR data. Error bars show statistical errors as obtained from the *Musrfit* program. **b** Temperature dependent comparison of the μSR local mean field, bulk magnetization $M$ measured by SQUID magnetometry, and XMCD switching statistics for ZFC (green markers) and FH-FC (red markers). The pictographs illustrate a muon stopping in FiM clusters embedded in a PM sea. As the clusters grow in size at lower temperatures, the average internal B-field shows a negative shift indicated by the arrow in (**a**). The FiM switching statistical error margins are binomial confidence intervals (obtaining $x$-switches in $n$ trials gives a $\sqrt{(x/n*(1-x/n)/n)}$ error). The statistically most relevant FiM switching, summarized from three independent beamtimes from fresh samples, was measured at 5 K (ZFC), that is why the smallest error bar in FiM switching probability is achieved under these measurement conditions.

sample volume becomes magnetic, i.e., a long range magnetic order is established. The combination of bulk magnetization and LE-μSR data confirms the presence of a net magnetic moment imposed by the FiM order below $T_{c1} \approx 100$ K, followed by a spin-glass transition below $T_{c2} \approx 40$ K, where the coupling between the FiM clusters becomes strong enough for the sample to undergo a transition to a cluster spin-glass long range order.

The combined set of experimental and theoretical results allows us to rationalize the MSR switching mechanism. For a broader perspective Fig. 6b compares the temperature dependence of μSR local mean field, bulk magnetization, and XMCD switching statistics. As expected with the onset of the long-range order, the spin-freezing temperature $T_{c2}$ marks a minimum rather than an onset in FiM switching probability. Strikingly, above $T_{c2}$, the switching events dramatically increase along with the decreasing size of the FiM clusters, as highlighted by the red shaded region. This trend appears to be independent on whether the system was magnetized under ZFC or field-heated field-cooling (FH-FC) indicated by green and red markers, respectively. Moreover, at $T = 65$ K the XMCD asymmetry sometimes reaches values as high as 60% in contrast to the expected increase of magnetic disorder at elevated temperatures (Supplementary Fig. 4). This is reminiscent of the spontaneous thermally induced magnetic order in elemental spin glasses[45], which in our case begins to morph between individual FiM clusters as indicated by pictographs in Supplementary Fig. 4b. The magnetic reordering on a mesoscopic scale and the magnetic frustration highlighted in the theory section are therefore related to the glassy behavior. Importantly, our XMCD results show that the FiM switching with nanoampere currents is collective over macroscopic dimensions. This indicates that the magnetic order has to be correlated across local magnetic interactions on which the magnetic spin textures spread like a collective excitation through the sample. This is exactly what is expected for a correlated cluster spin-glass[46,47]. Thus, our experimental results illustrate efficient collective spin dynamics in a correlated spin glass.

In conclusion, the multiferroic $Ge_{1-x}Mn_x$Te system presents an outstanding platform for a great variety of complex intriguing physical phenomena which we decipher using an holistic approach: XMCD and theory confirm a bistable ferrimagnetic ground state, whereas the collective switching of this state in XMCD, together with μSR and magnetometry, indicate that this switching shapes up inside a cluster spin glass. For such systems we have demonstrated a unique new concept to harness magnetostochastic switching, which allowed us to describe novel magnetization dynamics. Under this MSR drive the current needed for switching of the magnetic state is reduced by many orders of magnitude compared to typical spin-orbit torque systems, allowing for the design of energy efficient non-volatile electronics. In such a design, the synchrotron radiation induced currents would be replaced by electric currents generated from electrical contacts or by using a laser source. On a more fundamental level, our results open up new avenues for studying collective spin dynamics in correlated spin glass systems.

## Methods
### Sample preparation
We have grown 200 nm $Ge_{0.87}Mn_{0.13}$Te thin films on InP(111) and $BaF_2$(111) substrates by molecular beam epitaxy. The choice for 13% Mn-doping is that for such a doping the Zeeman splitting is saturating[6]. A protective stack of amorphous Te- and Se-capping layers with a total thickness of ≈20 nm was used to avoid surface oxidation and degradation. Before mounting into the XTREME beamline sample holder[48], it was completely removed in a ultrahigh vacuum chamber by annealing the samples for 30–45 min at 250 °C; and recapped with ≈2 nm amorphous carbon by carbon-thread evaporation within the same vacuum chamber.

## Experimental methods

The XMCD setup at the XTREME beamline[48] allows us to measure magnetic moments from the averaged orbital and spin contribution projected along the out-of-plane magnetization **M**, which is collinear to the ferroelectric polarization **P** along the [111] direction[6]. Raw TEY and TFY signals measured with Keithley 428 current amplifiers were normalized with the $I_0$ photocurrent from the XTREME beamline refocussing mirror. All the signals were sampled from the current amplifier analog outputs, and further processed with a software control system to synchronize the on-the-fly scan for absorption measurements with the undulator[40]. All XMCD spectra, except those in Supplementary Fig. 1, were obtained by taking the difference of ($\mu$+ - $\mu$-), where $\mu$+ and $\mu$- represent the XAS spectra measured by TEY, or TFY, with right- and left-handed circular light polarisation. The XMCD measurements were recorded as on-the-fly scans for a given magnetic field up to 6 T either parallel or anti-parallel to the beam direction. All data presented were measured during three independent beamtimes on fresh samples prepared in exactly the same way. The same samples were used for further NEXAFS (SuperXAS beamline at Swiss Light Source) and SQUID magnetometry.

Special extra-large samples capped with Au were prepared for low energy muon spin relaxation experiment (LE-$\mu$SR) at the $\mu$E4 muon beamline at the Paul Scherrer Institut[49]. LE-$\mu$SR time spectra are measured with two positron detectors placed opposite to each other for the considered samples. Temperature dependent asymmetry spectra were obtained under an external magnetic field of 5 mT applied perpendicular to the initial muon spin polarization direction. The data indicate magnetically ordered islands with a net magnetic moment aligned with the applied field, embedded in a sea of paramagnetic environment. As these islands grow in size, their magnetic moment along the applied field increases and the strength of their dipolar field in the paramagnetic regions increases. This dipolar field is primarily opposite to the direction of the applied field and hence produces the observed negative shift in $B(T)$ as seen in Fig. 6a. At the lowest temperatures, the remaining 5 mT oscillation is not due to the sample. Instead, it originates from the background contribution[50].

In order to achieve comparable local probe conditions with the XMCD photoelectron escape depth ($\approx$10 nm), the muon implantation energy was adjusted to set their stopping distribution just below the sample surface. Under magnetization, the asymmetry of the transverse field the muon precession decays. The main quantity determining the LE-$\mu$SR spectra in Fig. 6a is the muon spin precession amplitude decay from asymmetric emission of positrons, analyzed with *Musrfit* program[51].

## Crystal structure and computational details

Using first-principles calculations within the density functional theory, we have investigated the ground state magnetic properties of bulk $Ge_{0.867}Mn_{0.133}Te$. The calculations were carried out using the multiple scattering KKR Green function method as implemented in the spin-polarized fully relativistic Korringa-Kohn-Rostoker (SPRKKR) code[52]. The exchange and correlation effects were incorporated within the LDA framework[53]. Brillouin zone integrations were performed on a $39 \times 39 \times 39$ dense mesh of $k$-points. The angular momentum expansion up to $l_{max} = 4$ has been used for each atom. The energy convergence criterion and coherent potential approximation (CPA) tolerance has been set to $10^{-7}$ Ry. The potential is constructed in full potential geometry. The influence of chemical disorder in $Ge_{0.867}Mn_{0.133}Te$ can be estimated by means of the CPA alloy theory[54,55] implemented in the SPR-KKR method.

We consider two types of $Mn_i$ atoms denoted as $Mn_2$ and $Mn_3$ in Fig. 2a, which occupy allowed Wyckoff positions within the $Ge_{0.87}Mn_{0.13}Te$ space group (see Table 1), derived from the experimental data as determined from the room temperature X-ray powder diffraction measurements[56].

**Table 1 | $Ge_{0.867}Mn_{0.133}Te$ R3mR (No. 160) space group crystal structure with atomic positions and occupancies**

| $Ge_{0.867}Mn_{0.133}Te$ | |
|---|---|
| space group: R3mR (No. 160), $a$ = 4.234Å, $\alpha$ = 59.07° | |
| Wyckoff sites with atomic positions | Atomic occupancies |
| 1a (0.534,0.534,0.534) | $0.911Ge + 0.089Mn_s$ |
| 1a (0.0,0.0,0.0) | Te |
| 1a (0.250, 0.250, 0.250) | $0.978E_s + 0.022Mn_2$ |
| 1a (0.764, 0.764, 0.764) | $0.978E_s + 0.022Mn_3$ |

$E_s$ represents empty spheres used in the CPA formalism, $Mn_i$ with two atomic positions ($Mn_2$,$Mn_3$) occupying allowed Wyckoff positions of R3mR space group. The atomic distribution of Mn atoms as substitutional ($Mn_s$; 67%) and interstitial ($Mn_2$,$Mn_3$; 33%) were obtained from NEXAFS measurements summarised in Supplementary Fig. 1c.

**Table 2 | Spin ($m_S$) and orbital ($m_L$) magnetic moments obtained from CPA theory vs XMCD**

| CPA theory $\mu_B$/atom | | | | XMCD ($Mn_s$+$Mn_i$) $\mu_B$/atom |
|---|---|---|---|---|
| | $Mn_s$ | $Mn_i$ | $\Sigma Mn_{s,i}$ | MSR-off 0 T |
| $m_S$ | 4.209 | −2.112 | 2.097 | 0.093 ± 0.016 |
| $m_L$ | 0.047 | −0.079 | −0.032 | 0.040 ± 0.004 |

$m_S$ and $m_L$ momenta were calculated along the $Ge_{0.87}Mn_{0.13}Te$[111] easy magnetization axis. Besides dominant Mn $d$ states, theoretical values include small contributions from the $s$ and $p$ valence orbitals. On the other hand, due to dipole selection rules, XMCD data consider only Mn $3d$ contributions.

In addition we also performed $2 \times 2 \times 2$ super-cell Quantum ESPRESSO pseudopotential relaxation calculations with the Mn atomic positions summarized in Table 1. The results confirmed that AFM is the energetically most stable configuration. The structural models from the super-cell calculations were used to model XAS and XMCD calculations, which in turn where consistent with our CPA model summarized in Fig. 2d. Moreover, the structural relaxations around the Mn atoms showed negligible impact on the magnetic structure and shape of the XAS spectra.

Finally, we also performed multiplet calculations based on localized $Mn^{2+}3d$ with parameters reported by Sato et al.[17]. The comparison with measured $Ge_{0.87}Mn_{0.13}Te$ XAS spectra in Supplementary Fig. 1c and Fig. 2a confirms that our XAS and XMCD spectra feature localized $Mn^{2+}3d$ states. Within the SPRKKR multiple scattering methodology we consider an average of 4.72 holes for $Mn_s$ and 3.73 holes for $Mn_i$ unoccupied d states with a correction factor of 1.47 for compensating the jj-mixing[57]. However, as reported by Sato et al.[17], multiplet calculations suffer from intensity deviations between the calculations and experiments. For a better comparison with experimental data we thus relied on CPA calculations.

Because the CPA enables us to address spectral contributions from individual Mn-atoms, we tentatively modelled the resulting magnetic dichroism as a simple sum of all Mn contributions in saturation, as shown in Supplementary Fig. 2d. When reducing the $Mn_s$ spectral contribution by a factor of 3, we find that the resulting XMCD signal (dashed line in Fig. 2d) reproduce the measured data very well, whereas multiplet calculations with $Mn_s^{2+}$ atoms fail to reproduce the exact shape of the dichroism effect and lead to a spectral shift indicated by the black arrow (see Supplementary Fig. 2d). The factor 3 reduction of the $Mn_s$ spectral contribution is rationalized by observing a *significant decrease* in $Mn_s$ dichroism between the states right after the FH-FC@6T at 10 K, and recorded later during MSR-on conditions (vertical arrow in Supplementary Fig. 2b). With every FiM switching event at low temperatures some of the $Mn_s$ becomes part of the paramagnetic spin glass background; and when around one third of the original dichroism effect is left, this process appears to saturate

(Supplementary Fig. 2c). However, on increasing the temperature from $10 \to 45\text{-}65$ K, we observe a *significant increase* in the $Mn_s$ dichroism, which we attribute to self-induced magnetization stimulated by MSR along with the paramagnetic spin glass background as schematically depicted in Supplementary Fig. 4b.

## Magnetic ground state calculations

Figure 2 a is a cartoon description of various Mn-Mn exchange interactions between individual $S_1$, $S_2$ spin states considered in our model Hamiltonian, where Mn impurities act at the same time as random magnetic moments and as acceptors producing the carriers. For the $Ge_{0.87}Mn_{0.13}Te$ magnetic ground state we consider the general spin Hamiltonian in the following form:

$$H = -\frac{1}{2}\sum_{ij} J_{i,j} \mathbf{S}_i \cdot \mathbf{S}_j + \sum_{ij} \mathbf{D}_{i,j} \cdot (\mathbf{S}_i \times \mathbf{S}_j) + \sum_i K_i \mathbf{S}_i^2 \quad (1)$$

The first term represents the Heisenberg exchange energy with $\mathbf{S}_i$ and $\mathbf{S}_j$ unit vectors having directions corresponding to local magnetic moments on sites $i$ and $j$. $J_{i,j} > 0$ and $J_{i,j} < 0$ prefer FM and AFM spin configurations, respectively. The second term originates from the antisymmetric part of the interaction matrix, also termed as antisymmetric exchange or Dzyaloshinskii-Moriya interaction (DMI). Finally the $K_i$ term in the final term accounts for the magnetocrystalline anisotropy energy (MAE).

The isotropic $J_{i,j}$ exchange between localized $S_1$, $S_2$ spin states depends on the distance between the next-nearest Mn dopants. In contrast to earlier theoretical $Ge_{1-x}Mn_xTe$ studies considering only $Mn_s$ atoms[20], another type of interaction is coming from the $\sum_{i,j} \mathbf{D}_{i,j} \cdot (\mathbf{S}_i \times \mathbf{S}_j)$ Dzyaloshinskii-Moriya interaction obtained by sampling the magnetization deflection between [111] and [$\overline{111}$] easy magnetization axis directions. Below we discuss the contributions of individual terms to the $Ge_{0.87}Mn_{0.13}Te$ magnetic ground state.

**Isotropic exchange Interactions.** The magnetic exchange coupling parameters ($J_{i,j}$) are based on the real space approach by using the theory proposed by Liechtenstein et al.[58]. This approach employs the "magnetic force theorem" to determine $J_{i,j}$ by assessing the total energy change related to an infinitesimal rotation of the magnetic moments located at the atomic sites $i$ and $j$. The energy change can be related to the exchange coupling parameters $J_{i,j}$ as:

$$J_{i,j} = \frac{1}{4\pi} \int^{E_F} dE \, \mathrm{Im} \, Tr_L \{\Delta_i \tau_\uparrow^{i,j} \Delta_j \tau_\downarrow^{j,i}\} \quad (2)$$

where $\tau$ is the scattering path operator, $\Delta_i$ is the difference in the inverse single site scattering $t$ matrices for spin up and spin down electrons, and $Tr_L$ is the trace of scattering matrices over the orbital indices $L = (l, m)$.

The $J_{i,j}$ calculations are performed within a cluster of radius $3a$, where $a$ is the lattice parameter. We neglect all interactions involving Ge and Te atoms and consider only those between Mn atoms which are found to host significant localized magnetic moments/atom. Figure 2a of the main text shows three different types of Mn atoms inside the $Ge_{0.867}Mn_{0.133}Te$ primitive unit cell, classified as substitutional $Mn_s$ and interstitial $Mn_i$ atoms, respectively.

The physical mechanism behind the magnetic ground state has been attributed to *pd*-exchange coupling[20,59], combined with *d-d* magnetic interactions[17]. Consistently with our theoretical predictions, the *pd*-exchange coupling is relatively weak, but long ranged. Moreover, for $Ge_{1-x}Mn_xTe$ the *pd*-exchange constant is negative[17], stemming from AFM exchange interaction between $Mn^{2+}$ states. In agreement with our $Ge_{1-x}Mn_xTe$ magnetic ground state description, this negative term naturally materializes between $Mn_i$-$Mn_s$ which further confirms

that $Ge_{1-x}Mn_xTe$ is a ferrimagnetic rather than a ferromagnetic DMS[16-20].

**Dzyaloshinskii-Moriya interaction.** The DMI is an anisotropic chiral exchange interaction between localized spins and has a net contribution only in systems without structural inversion symmetry and with the presence of an indirect or long-range exchange interaction. $Ge_{0.87}Mn_{0.13}Te$ satisfies these two conditions, which is also evident from the isotropic exchange interactions described in the previous section. The origin of DMI is found in the spin-orbit coupling (SOC) which acts as a perturbation on localized orbital states. Given two neighboring spins $\mathbf{S}_1$ and $\mathbf{S}_2$, the DMI energy can be described by $-\mathbf{D}_{1,2} \cdot (\mathbf{S}_1 \times \mathbf{S}_2)$. Therefore, the direction of the relative $\mathbf{S}_1$ and $\mathbf{S}_2$ rotations can be clockwise or counter-clockwise, providing the information about helicity in the case of spin spirals. This expression is part of a generalized exchange interaction where the DMI term is related to the exchange constant $J$ of the direct Heisenberg exchange $-J(\mathbf{S}_i \cdot \mathbf{S}_j)$. However, contrary to the latter which favors collinear alignment, the DMI promotes an orthogonal arrangement between $\mathbf{S}_i$ and $\mathbf{S}_j$, with a chirality imposed by the direction of $\mathbf{D}_{i,j}$. The resulting spin helicity is uniform, meaning that clockwise or counterclockwise rotation of spins are energetically identical. Data in Fig. 2c show all the components of $\mathbf{D}_{i,j}$ (i.e. $D_{i,j,x}, D_{i,j,y}, D_{i,j,z}$) interaction, showing robust exchange up to a cluster size of $3.0a$. The average DMI energy is comparable to the $J_{i,j}$ exchange energy for larger distances; and a factor 4 higher than the magnetic anisotropy energy (MAE) discussed below.

The effective ratio between the exchange interaction and DMI for a given pair $i, j$ was calculated in the mean field approach as

$$\left(\frac{J}{D}\right)_{\mathrm{eff}} = \sum \left( C_i C_j \frac{J_{i,j}}{D_{i,j}} \right) \quad (3)$$

where the summation runs over all the neighbours up to a cluster of radius $3.5a$ where $a$ is the lattice parameter. The factors $C_i$ and $C_j$ represent the occupation of the Mn atom at a given site. This yields 19.933, 20.7919, and 2.4275 for the Mn pairs $i, j = 1, 1$, $1, 2$, and $2, 3$, respectively.

**Magnetic anisotropy energy.** The magnetic ground state of FiM exchange energy (grey markers in Fig. 3a) has been explored by determining the total energy corresponding to the relative spin orientation of $Mn_2$ and $Mn_3$ spins with respect to the substitutional $Mn_1$ spins. The $Mn_1$ spin was frozen along the [111] quantization axis and the $Mn_2$, $Mn_3$ spins were rotated along the out-of-plane [111] direction starting from $\theta = 0$, which corresponds to the FM state with all spins pointing in the same direction. However, the self consistency convergence in calculations can be achieved only for a limited $\theta$ range between $40°\text{-}320°$. Nevertheless, the total energy of the system reaches a minimum with $Mn_1$ spins set to $0°$ and $Mn_{2,3}$ spins to $180°$, which confirms the FiM order. The red markers in Fig. 3a show the MAE energy variation as a function of magnetization angle $\theta$, evaluated as a difference between the fully relativistic total energies calculated for quantization axes [111] and axis orthogonal to [1$\overline{1}$1]. Both FiM exchange and MAE consider a magnetoelectric ground state with ferroelectric distortion and the easy magnetization axis alignment along the [111] or [$\overline{111}$] directions. The calculated MAE of the $Ge_{0.867}Mn_{0.133}Te$ is 0.101 meV/f.u. and represents the energy barrier between the easy and hard magnetic axes. Finally, the energy landscape obtained by combination of MAE and FiM ground state was used to model the FiM double-well potential as depicted in Fig. 3a. The MSR-driven FiM switching contracts the states into one of the two wells. The bistable potential depicted can be expressed as $U(x) = -\frac{1}{2}ax^2 + \frac{1}{4}bx^4$ with minima located at $\pm 90°$ and barrier height $\Delta V = a^2/4b$.

## Data availability

The data that support the findings of this study are available from the corresponding authors upon request.

## Code availability

In the manuscript we used two ab-initio DFT based packages. SPRKKR multiple scattering package is freely available (no costs apply) under the specific user license and the package can be downloaded upon registration (https://www.ebert.cup.uni-muenchen.de/index.php/en/software-en). The Python based interface, *ase2sprkr*, is published under the MIT-license (https://github.com/ase2sprkr/). Quantum ESPRESSO suite can be downloaded under GNU General Public License and is open source code (https://www.quantum-espresso.org/). All post-processing procedures and scripts used to evaluate experimental and theoretical data are available from the authors upon request.

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

## Acknowledgements
We thank Stefano Rusponi and Valerio Scagnoli for fruitful discussions. J.H.D. gratefully acknowledges financial support from the Swiss National Science Foundation project No. PP00P2_170591. J.M. and S.dS. thank the CEDAMNF with reg. no. CZ.02.1.01/0.0/0.0/15_003/0000358 and within the project QM4ST with reg. no. CZ.02.01.01/00/22_008/0004572, co-funded by the ERDF as part of the MŠMT. G.S. acknowledges support by the Austrian Science Funds (FWF), Projects P30960-N27 and I4493-N. DK acknowledges the Lumina Quaeruntur fellowship of the Czech Academy of Sciences. M.G. acknowledges the financial support under grants APVV-SK-CZ-RD-21-0114, VEGA 1/0105/20 and Slovak Academy of Sciences project IMPULZ IM-2021-42 and project FLAG ERA JTC 2021 2DSOTECH.

## Author contributions
G.S., J.K., J.M., and J.H.D. initiated and coordinated the project on equal level; S.W.DS. performed the main calculations under supervision of J.M.; supporting calculations were carried out by M.G.; XMCD measurements: J.K, C.P., J.H.D., M.F., C.A.F.V.; LE-$\mu$SR experiments: J.K., J.A.K., Z.S. and T.P.; NEXAFS measurements: A.N., D.K., J.K.; SQUID measurements and magnetometry: A.N., J.K.; data analysis: J.K., A.N., O.C. J.H.D. ; writing of manuscript: J.K., J.M., G.S., J.H.D. All authors extensively discussed the results and the manuscript.

## Competing interests
The authors declare no competing interests.
