## [Peer Review File · Nature Communications]

Reviewers' Comments:

Reviewer #1:

Remarks to the Author:

I would like to reiterate that this study presents an intriguing result. In the revised version of the manuscript, the authors have appropriately addressed my previous concerns by removing unsupported arguments, such as the discussion on topological spin textures. Additionally, all of my other comments have been adequately resolved.

Overall, I have no doubt that it will attract interest from diverse communities, including spintronics and ferroelectrics. Furthermore, this study is expected to stimulate further research in the fields. Therefore, I highly recommend its publication in *Nature Communications*.

Reviewer #2:

Remarks to the Author:

I reviewed the author's response as well as the revised manuscript. The authors made substantial revisions to the manuscript resulting in a more robust report focusing on the interesting experimental observation of energy efficient MSR switching of magnetic state in cluster spin glass $\text{Ge}_{0.87}\text{Mn}_{0.13}\text{Te}$.

I would encourage the authors to consider a revised title that better captures the unique results presented here, such as "Efficient magnetostochastic resonance switching of magnetic state in a correlated spin glass" since "collective spin dynamics" does not really capture the essence of the report. This is simply a suggestion and the authors can choose to neglect it.

I can recommend publication of this paper in *Nature Communications* after the authors make revisions to address the following points:

1) Ref. 39 is an added note with estimated pulse current density. There seems to be an error in the unit conversion from mm^2 to cm^2 . A current of 100 nA in an area of 1 mm^2 is $100 \times 10^{-9} / (1\text{mm} \times 1\text{mm}) = 10^{-7} / (0.1\text{cm} \times 0.1\text{cm}) = 10^{-5} \text{A}/\text{cm}^2$ or two orders of magnitude smaller than what is stated. Various statements along the paper relating to the reduced pulsed current densities need to be adjusted.

2) Magnetic moment is $M = m_L + m_s$, with $m_s = -2S_z$. It seems that the authors are mixing definition of magnetic moment and spin angular momentum by reporting total moment to be $M = m_L + 2m_s$. Table II of Methods III appears to report $-S_z$ values instead of m_s . It begs the question, what are the values reported in the same table for the CPA results, m_s or $-S_z$? It looks like Figure 5 reports m_s ($-2S_z$) while Table II of Methods III reports $-S_z$. This needs to be checked and clarified/adjusted in text and figures.

3) The comparison of derived moments from XMCD to those in Ref. 43 seems problematic. Ref. 43 assumes that the local moment is 5 $\mu\text{B}/\text{Mn}$ and assigns an x value to the measured magnetization based on that assumption. For example, a value of 0.5 $\mu\text{B}/\text{f.u}$ corresponds to $x=0.1$. The values reported in Ref. 43 for $x=0.13$ are around 0.6 $\mu\text{B}/\text{f.u}$ (or per 0.13 Mn) which corresponds to $0.6/0.13 \mu\text{B}/\text{Mn} =$ or about 4.6 $\mu\text{B}/\text{Mn}$. The statement that the XMCD-derived value of 0.17 $\mu\text{B}/\text{Mn}$ is in agreement with Ref. 43 is misleading.

REVIEWER COMMENTS

Reviewer #1 (Remarks to the Author):

I would like to reiterate that this study presents an intriguing result. In the revised version of the manuscript, the authors have appropriately addressed my previous concerns by removing unsupported arguments, such as the discussion on topological spin textures. Additionally, all of my other comments have been adequately resolved.

Overall, I have no doubt that it will attract interest from diverse communities, including spintronics and ferroelectrics. Furthermore, this study is expected to stimulate further research in the fields. Therefore, I highly recommend its publication in Nature Communications.

Reply: We thank the referee for the positive evaluation of the revised manuscript and for highly recommending publication.

Reviewer #2 (Remarks to the Author):

I reviewed the author's response as well as the revised manuscript. The authors made substantial revisions to the manuscript resulting in a more robust report focusing on the interesting experimental observation of energy efficient MSR switching of magnetic state in cluster spin glass $\text{Ge}_{0.87}\text{Mn}_{0.13}\text{Te}$.

I would encourage the authors to consider a revised title that better captures the unique results presented here, such as "Efficient magnetostochastic resonance switching of magnetic state in a correlated spin glass" since "collective spin dynamics" does not really capture the essence of the report. This is simply a suggestion and the authors can choose to neglect it. I can recommend publication of this paper in Nature Communications after the authors make revisions to address the following points:

We thank the reviewer for carefully reading our revised manuscript and for recommending publication! As suggested, and in order to highlight the essence of the manuscript we simplified the title as follows:

"Efficient magnetic switching in a correlated spin glass"

1) Ref. 39 is an added note with estimated pulse current density. There seems to be an error in the unit conversion from mm^2 to cm^2 . A current of 100 nA in an area of 1 mm^2 is $100 \times 10^{-9} / (1 \text{ mm} \times 1 \text{ mm}) = 10^{-7} / (0.1 \text{ cm} \times 0.1 \text{ cm}) = 10^{-5} \text{ A/cm}^2$ or two orders of magnitude smaller than what is stated. Various statements along the paper relating to the reduced pulsed current densities need to be adjusted.

We thank the reviewer for noticing this error. The note in Ref. 39 is now corrected. Moreover, as suggested by the referee, in the main text we now relate pulsed current densities instead of pulsed currents. Therefore, we replaced:

..., i.e. with current pulses six orders of magnitude lower compared to typical spin-orbit torque systems [39]

With:

with unprecedented efficiency of low-current density pulses compared to typical spin-orbit torque systems [39]

2) Magnetic moment is $M = m_L + m_s$, with $m_s = -2S_z$. It seems that the authors are mixing definition of magnetic moment and spin angular momentum by reporting total moment to be $M = m_L + 2m_s$.

We thank the reviewer for pointing out this mistake in the definition. Of course, m_s refers to spin magnetic moment. The total moment M_{tot} in main text is corrected to $M_{\text{tot}} = m_s + m_L$

Table II of Methods III appears to report $-S_z$ values instead of m_s . It begs the question, what are the values reported in the same table for the CPA results, m_s or $-S_z$? It looks like Figure 5 reports m_s ($-2S_z$) while Table II of Methods III reports $-S_z$.

The spin and orbital magnetic moments as calculated by CPA reported in Table II are in units of μ_B/atom . In addition, in calculating the spin magnetic moment from the sum rules we corrected the number of holes in the Mn unoccupied d-states as implemented in the SPR-KKR method. Because (Ge,Mn)Te with 13% of Mn consists of substitutional (Mn_s) and interstitial atoms (Mn_i), the effective number of holes is an average of 4.72 holes for Mn_s and 3.73 holes for Mn_i .

As a correction, in Methods this sentence:

Therefore, for the XMCD sum rule calculations we considered 5 holes in the Mn unoccupied d-states with a correction factor of 1.47 for compensating the jj-mixing [59].

is replaced as follows:

Within the SPRKKR multiple scattering methodology we consider and average of 4.72 holes for Mn_s and 3.73 holes for Mn_i unoccupied d-states with a correction factor of 1.47 for compensating the jj-mixing [59].

Based on these corrections, the $M_{\text{tot}} = m_s + m_L$ yields $0.13 \pm 0.1 \mu_B/\text{Mn atom}$ instead of $0.17 \pm 0.1 \mu_B/\text{Mn atom}$.

This needs to be checked and clarified/fixed in text and figures.

The reconsidered values for the spin magnetic moment were fixed in the main text, Methods and Fig.5a. In addition, the caption of Table II is specifying the difference between CPA and XMCD results by adding this sentence:

Besides dominant Mn d states, theoretical values include small contributions from the s and p valence orbitals. On the other hand, due to dipole selection rules, XMCD data maps only Mn3d contributions.

3) The comparison of derived moments from XMCD to those in Ref. 43 seems problematic. Ref. 43 assumes that the local moment is 5 μ_B /Mn and assigns an x value to the measured magnetization based on that assumption. For example, a value of 0.5 μ_B /f.u corresponds to $x=0.1$. The values reported in Ref. 43 for $x=0.13$ are around 0.6 μ_B /f.u (or per 0.13 Mn) which corresponds to $0.6/0.13 \mu_B/\text{Mn}$ or about 4.6 μ_B/Mn . The statement that the XMCD-derived value of 0.17 μ_B/Mn is in agreement with Ref. 43 is misleading

We thank the reviewer for raising this question. Indeed, Ref.43 assumes that the local moment is 5 μ_B /Mn atom. They assign the measured magnetization based on that assumption because they consider the system to be ferromagnetic. As explained in the “Magnetic ground state properties” section, (Ge,Mn)Te, with 13% of Mn is a diluted ferrimagnetic system. We agree with the reviewer that under these assumptions a comparison can be misleading. To avoid confusion, we thus removed this sentence from the main text:

However, the obtained values are in good agreement with earlier SQUID studies on $\text{Ge}_{1-x}\text{Mn}_x\text{Te}$ crystals, which yielded 0.17-0.2 μ_B /f.u. [43].

Because Ref.43 is an important work providing SQUID magnetometry, we relocated it into the introductory section on “Magnetic ground state properties”.

Again, we would like to thank the reviewer for rigorous inspection of our manuscript! We would like to highlight that, although the corrections made are important for future reference, they do not change the findings or interpretation of the manuscript.